# Increased Utilization of Abdominal Surgical Procedures, Endoscopy and Imaging After Negative Rectal Biopsies for Suspected Hirschsprung’s Disease: A Danish Nationwide Matched Cohort Study

**DOI:** 10.3390/children12091112

**Published:** 2025-08-24

**Authors:** Niels Bjørn, Gunvor Madsen, Rasmus Gaardskær Nielsen, Jonas Sanberg, Niels Qvist, Mark Bremholm Ellebæk

**Affiliations:** 1Research Unit for Surgery, Odense University Hospital, University of Southern Denmark, 5230 Odense, Denmark; niels.bjorn@rsyd.dk (N.B.); mark.ellebaek1@rsyd.dk (M.B.E.); 2Centre of Excellence in Gastrointestinal Diseases and Malformations in Infancy and Childhood (GAIN), Odense University Hospital, University of Southern Denmark, 5230 Odense, Denmark; 3Odense Patient Data Explorative Network (OPEN), Odense University Hospital, 5230 Odense, Denmark; 4Research Unit for Pathology, Odense University Hospital, University of Southern Denmark, 5230 Odense, Denmark; gunvor.madsen@rsyd.dk; 5Research Unit of Pediatrics, Odense University Hospital, University of Southern Denmark, 5230 Odense, Denmark; rasmus.gaardskaer.nielsen@rsyd.dk

**Keywords:** Hirschsprung’s disease, rectal biopsy, surgery, endoscopy, imaging, registry study

## Abstract

**Objective**: Functional constipation affects up to one third of children. When standard treatments fail, the child may be referred to rectal biopsy on the suspicion of Hirschsprung’s disease (HD), but only 7–28% will have a confirmed HD. For those with a negative biopsy for HD, there is limited evidence of the post-biopsy utilization of healthcare services. This study aimed to investigate gastrointestinal-related healthcare utilization after rectal biopsy in non-HD children compared to matched healthy controls and patients diagnosed with HD. **Methods**: This nationwide registry-based cohort study included all Danish children <18 years who underwent rectal biopsy for HD during the period from 1998 to 2018. The cohort was matched 1:10 to form a control cohort. Outcomes included gastrointestinal surgeries, endoscopies, and imaging procedures pre- and post-biopsy. **Results**: Among the 1105 children included in the cohort, 128 were diagnosed with Hirschsprung’s disease (HD), while 977 were non-HD. Compared to the control group, the non-HD showed significantly higher rates of post-biopsy surgery (11.2% vs. 1.6%, *p* < 0.001), endoscopy (9.4% vs. 0.5%, *p* < 0.001), and imaging (37.1% vs. 7.8%, *p* < 0.001) related to the gastrointestinal tract. **Conclusions**: Children with a HD-negative rectal biopsy had 5–10 fold increased frequency of gastrointestinal-related surgeries, endoscopies or imaging during the follow-up period compared to the background population.

## 1. Introduction

Functional constipation in infancy and childhood is a common condition affecting between 0.7% and 29.6% of children, dependent upon the definitions and population studied [1]. The primary treatment is laxatives and behavioral adjustments. Those who fail treatment may be recommended a rectal biopsy on the suspicion of Hirschsprung’s disease (HD). A history of delayed passage of meconium, constipation since the first few weeks of life, chronic abdominal distension, vomiting, family history of Hirschsprung’s disease or failure to thrive strengthen the indications [2].

The prevalence of HD in Europe is approximately 1 in 5000 newborns [3]. The treatment for HD is surgical resection of the affected, aganglionic bowel segment. Lifelong bowel dysfunction is common, and planned follow-up care is recommended [4]. Despite clear clinical indications, only 7–28% of children referred for rectal biopsy are ultimately diagnosed with HD [5,6,7]. Children with normal histopathological findings (non-HD patients) may continue to experience functional bowel disorders, potentially leading to increased healthcare utilization; however, data on long-term outcomes in this group remain limited. Two previous studies have investigated outcomes related to constipation in biopsied non-HD children, with follow-up periods ranging from two to four years [8,9]. Both studies reported persistent constipation that was relatively more resistant to standard treatments compared to children with constipation who had not undergone biopsy. Additionally, the biopsied group had a higher rate of hospital admissions and was more often diagnosed with gastrointestinal-related conditions compared to controls. Childhood constipation is known to have a substantial impact on quality of life [10], with increased healthcare costs up to threefold higher than the background population, comparable to other pediatric conditions such as asthma and attention-deficit/hyperactivity disorder (ADHD) [11].

There have been numerous papers published on investigations performed in children with suspicion on HD. To our knowledge, no studies have investigated the use of radiological investigations, endoscopic procedures and surgical interventions related to the gastrointestinal tract in children who had undergone a rectal biopsy on the suspicion of HD without confirmed diagnosis. We hypothesized that these children would undergo more investigations and surgical procedures related to the gastrointestinal tract compared to a matched control group.

This study aimed to characterize the demographics of a Danish cohort of infants and children who underwent rectal biopsy due to suspected HD. The primary objective was to describe the utilization of gastrointestinal-related procedures before and after biopsy, including surgery, endoscopy and imaging, compared to the background population.

## 2. Materials and Methods

### 2.1. Study Design

This study is a nationwide registry-based and matched controlled cohort study of all Danish children (age < 18 years) who underwent any type of rectal biopsy on suspicion of HD from 1 January 1998 to 31 December 2018. The date for the biopsy was set as the index date, and follow-up continued until 31 December 2018, or until the date of death or emigration from Denmark, whichever came first.

### 2.2. Setting

The Danish healthcare system is tax-funded and provides free access to universal healthcare services. All Danish citizens are registered with a unique civil registration number (CRN) issued at birth or upon immigration to Denmark. The Danish national registries can be linked using the CRN and allow for epidemiological studies of the entire population, comprising 5.96 million inhabitants on 1 January 2024 [12].

### 2.3. Population

The exposed population was identified using the Danish National Pathology Registry (PatoBank). It contains data on all tissue samples received for pathological examination and diagnosis in Denmark from 1997 and onwards [13]. We used the individual SNOMED-classification codes T68 (rectum), TX9600 (autonomic nerve system), P30610 (biopsy), and P30615 endoscopic biopsy to identify eligible patients. Duplicate records, identified by matching CRN numbers, were electronically removed. Patients with confirmed HD were censored for analysis of eventual later rectal biopsies. The identified pathology reports were manually reviewed to confirm the description of the presence or absence of ganglion cells, which is an essential criterion in histological evaluation of rectal biopsies performed on the suspicion of HD. A validation through a re-assessment process involving 20 randomly selected cases, which were reviewed twice after a 30-day interval, was performed to ensure consistency in the diagnosis of HD or non-HD.

The exposed cohort was divided into two subgroups: one consisting of patients with histologically confirmed diagnosis of HD in the Nations Patient Registry (ICD-10: DQ43), and a group without HD (Non-HD).

An unexposed cohort (control group) without any registered rectal biopsy was matched at a 10:1 ratio using data from the Danish Civil Registry [14]. The 10:1 matching was used because of the relatively low number of exposed individuals, and to increase statistical power. Matching included age at biopsy date and region of residence on the index date. Matching was performed on the whole group of exposed patients. The inclusion of region of residence was to account for potential regional differences in the indication for biopsy and the used method (full-thickness or suction).

All gastrointestinal-related surgeries (including abdominal wall surgery), endoscopies and radiological examinations were retrieved from the National Patient Registry (NPR). The NPR contains records of discharge diagnoses since 1977 and all outpatient diagnoses since 1994) [13,15]. From 1995 and onwards, the NPR has used the ICD-10 classification. All procedures, such as endoscopies, surgeries and radiological examinations, are registered using the Nomesco Classification of Surgical Procedures [16]. The complete list of retrieved procedural codes appears from Appendix A.

### 2.4. Outcomes

The frequency of surgeries, radiological examinations and endoscopic procedures related to the gastrointestinal tract before and after the index date were compared across the three cohorts (HD, Non-HD and controls).

### 2.5. Covariates

Based on the CRN, the Danish Civil Registry and Statistics Denmark maintains complete records of births, deaths, civil status, household income and emigration status of the entire Danish population. Perinatal data, including birth weight, APGAR score, parity and mother’s comorbidities (smoking status, alcohol consumption and obesity) were obtained from the Danish Medical Birth Registry, which was established in 1973. The register monitors pregnant women’s and their offspring’s health and provides data for quality assessment of perinatal care in Denmark [17].

### 2.6. Statistical Analysis

The Chi^2^ test or Fisher’s exact test was used for the analysis of categorical data, and the student’s t-test or Kruskal–Wallis test for continuous variables were used where applicable for both demographic variables and outcomes.

We used logistic regression analysis to evaluate the association between group (biopsy without HD vs. control) and the post-biopsy date using surgery, endoscopy, or imaging. Logistic regression was performed as a univariate analysis, and covariates with a significance of less than 0.05 were included in the multivariate analysis. Covariates included sex, birth length, gestational age, APGAR-score, mother’s BMI, mother’s smoking status, equivalated family income and surgery, endoscopy or imaging before biopsy/index date. In the logistic regression analyses, children diagnosed with HD were excluded due to collinearity.

Statistical significance was defined as a two-tailed *p*-value of less than 0.05. All data management and statistical analysis were performed using STATA 18 (College Station, TX, USA) on a secure server at Statistics Denmark.

### 2.7. Data Management

The processing of personal data was reported to and approved by the Region of Southern Denmark and is recorded in the internal open registry (Ref. 22/56234) in accordance with Article 30 of the EU General Data Protection Regulation (GDPR).

Patient registration numbers were extracted from the Patobank registry and transferred to the Danish Health Data Authority (*Sundhedsdatastyrelsen*). The data were subsequently anonymized, stored and processed exclusively within the OPEN (Odense Patient data Explorative Network) database, where all analyses were conducted. In compliance with Danish Statistics (DST) regulations, outcomes involving fewer than five patients will not be published.

## 3. Results

From a total of 4131 eligible patients, 1638 were excluded (all below 18 years of age) because the pathology report did not contain a wording of ganglion cells or HD. Of the remaining 2494, a total of 1389 were duplicates and were removed. Thus, 1105 patients were included in the exposed cohort. The final exposed cohorts thus consisted of 128 HD patients and 977 non-HD patients. The control group included 11,077 children (Figure 1).

In the HD group, a significantly higher proportion of males was observed compared to the non-HD group (Table 1). Children in the HD group were also younger at inclusion, with a mean age at the time of biopsy of 0.9 years (SD 2.0), compared to 2.8 years (SD 3.2) in the non-HD group (*p* < 0.001). The follow-up time was longer in the HD group, with a mean of 6.8 years (SD 3.9), compared to 5.6 years (SD 3.8) in the non-HD group (*p* = 0.002). In the non-HD group, gestational age, birth length, and APGAR scores after 5 min were significantly lower compared to the control group, and a higher proportion of mothers were smokers. No other significant differences between the groups were identified.

### 3.1. Surgery

Prior to biopsy, 16.2% of the children in the non-HD group had undergone surgery, compared to 1.2% in the control group (*p* < 0.001). After biopsy and during the follow-up, the figures were 11.2% and 1.6%, respectively, (*p* < 0.001) as shown in Table 2.

Multivariate regression analysis demonstrated that non-HD children had increased odds of undergoing surgical procedures after the index date compared to controls (OR: 4.93, 95% CI: 3.46–6.99, *p* < 0.001). Male sex (OR: 1.45, *p* = 0.004), maternal smoking during pregnancy (OR: 1.40, *p* = 0.027), and history of surgery before biopsy (OR: 2.65, *p* < 0.001) were significantly associated with an increased risk of surgery after biopsy (Appendix A).

The distribution of the type of surgical procedures during follow-up is presented in Table 3. In the non-HD group, surgeries involving the small and large intestine were the most common, comprising 39.4% of all procedures, followed by abdominal wall surgeries (20.8%) and ano-/rectal surgeries. In the control group, abdominal wall surgeries were the most common (58.7%), followed by surgery on the appendix, appendectomy (21.8%).

### 3.2. Endoscopy

Prior to biopsy, 6.2% of the non-HD children had undergone endoscopy compared to 0.3% of the controls (*p* < 0.001); after the index date, the rates were 9.4% and 0.5%, respectively (*p* < 0.001). The OR was 10.89 (95% CI: 6.80–17.45, *p* < 0.001), with prior endoscopy (OR: 4.21, *p* = 0.001) and prior imaging (OR: 1.70, *p* = 0.0026) as additional predictors (Appendix A).

### 3.3. Imaging

Prior to biopsy, 65.9% in the non-HD group had undergone abdominal imaging compared to 5.0% in the control group (*p* < 0.001). After biopsy, the figures were 37.1% and 7.8%, respectively (*p* < 0.001). The OR was 9.47 (95% CI: 6.89–13.01, *p* < 0.001). A history of imaging (OR: 1.93, *p* < 0.001) and prior endoscopy (OR: 2.23, *p* = 0.004) were significantly associated with higher imaging use after biopsy (Appendix A).

## 4. Discussion

The present study showed that children who had undergone a rectal biopsy on the suspicion of HD and without a confirmed diagnosis had approximately 10- and 5-times higher odds ratios of having abdominal surgery before and after biopsy, respectively, compared to the background population. For endoscopy and imaging, similar results were found. This is in line with the findings of Harlev et al., who found that children who had undergone rectal biopsy without a conformed HD were significantly more often hospitalized or diagnosed with a GI-related condition compared to controls [9]. In the study by Tran et al., children with severe constipation who had undergone a rectal biopsy for evaluation of HD had a less successful outcome compared to children with chronic constipation but without a rectal biopsy [8]. These results may reflect a more treatment-refractory phenotype of constipation in patients that are referred for rectal biopsy without a confirmed diagnosis of HD.

Anorectal manometry and contrast studies were not routinely part of the diagnostic work-up for Hirschsprung’s disease in Denmark during the study period or thereafter, in line with the NICE recommendations [2]. The high incidence of imaging procedures prior to the index data (biopsy) found in both the HD and non-HD group may reflect widespread use of diagnostic imaging prior to the decision of biopsy, but we have no information on the results of these investigations. Diagnosis relied primarily on rectal biopsy.

The most surprising finding might be that surgeries on the small and large bowel were the most frequent procedure, but the design of the present study does not reveal any information on indications or other diagnoses related to these interventions. It is also unknown whether this trend may persist beyond this relatively short observation period. In the control group, the prevailing surgical procedures were abdominal wall surgeries (hernia) and surgeries on the appendix (appendectomy). The procedure code specific for appendicostomy, which is an option for treating severe constipation [18], could not be retrieved from the dataset. Our search for surgical interventions was confined to the main groups, including the stomach, small or large bowels, etc., because the number of the detailed procedures performed was very low, and a statistical analysis would have no meaning. Also, we could not distinguish between acute or elective surgery.

Registry-based data inherently have limitations. They offer valuable insights into treatment patterns, but they do not provide detailed information on the indications for the specific treatments. As such, registry data cannot replace the granularity of prospective clinical data when it comes to causality or individualized patient analysis.

Thus, the pattern in surgeries suggests a gastrointestinal symptom burden in non-HD children that potentially increases the risk of having major surgery in the gastrointestinal tract. These findings emphasize the need for further investigation into diagnostic pathways, treatment decisions, and long-term management for children with severe constipation symptoms but without HD. A large registry-based study in pediatric populations with a diagnosis of constipation demonstrated significantly increased healthcare utilization [19,20]. As expected, the HD-group underwent more surgical procedures prior to the biopsy date, which reflects the necessity of acute surgical procedures in some patients before the diagnosis is confirmed histologically. However, it is remarkable that approximately half of patients may undergo four or more procedures. Whether this is due to complications or to disease/intervention-related conditions is unknown.

Abdominal wall surgeries, including hernia repairs, were included to capture the overall surgical burden in the non-HD group. While not directly caused by constipation, chronic straining and increased intra-abdominal pressure may contribute to hernia formation. Additionally, symptom overlap may lead to surgical evaluation. Colonic redundance is often reported in patients with chronic constipation without HD, but the clinical significance of this condition is unknown, and evidence for surgical resection does not exist.

The elevated proportion of endoscopy in the non-HD group compared to controls may reflect diagnostic efforts driven by persistent symptoms and may represent a self-perpetuating pattern of clinical surveillance. Regression analysis confirmed that prior endoscopy and imaging were strong predictors of subsequent endoscopy, although the drivers of increased utilization remain unclear.

Also, gastrointestinal imaging was more frequent in the non-HD group. Prior to biopsy, transabdominal ultrasound (US) was nearly tenfold more frequent compared to controls. After biopsy, US remained widely used in both the HD and non-HD groups. Although transabdominal rectal US is not recommended by NICE guidelines for constipation evaluation [2], it is frequently used in Denmark as a non-invasive method to assess rectal diameter to evaluate treatment of constipation [21]. Use of CT and MRI remained low across all groups. Although maternal smoking was significantly associated with increased rates of surgery, endoscopy, and imaging, it became insignificant after adjusting for gestational age, birth size, and prior procedures. It was not possible to perform a reliable sub-analysis due to the low number of cases.

A major strength of this study is the sample survey validation of histological results and the cross-verification of pathology reports with ICD-10 codes to ensure diagnostic accuracy of HD.

### Limitations

The pooling of HD and non-HD in the matching of the control group could be a confounder. However, the only significant differences between the HD and non-HD group were gestational age, maternal smoking, and APGAR scores; the numeric differences were small and not considered a serious confounder, with no major effects on the present results. There is also a risk that some patients in the HD group were missing, where the diagnosis was based on other investigations and not a rectal biopsy. The number was low and hence considered to not have a significant impact on the present results. We did not perform any review of the surgical specimens in patients that appeared in the NPR with a HD diagnosis without a rectal biopsy, which could be another limitation. Despite the generally high quality of Danish registries [15], the potential for coding errors exists; however, cross-referencing multiple data sources mitigated this risk. Although we could not distinguish between full-thickness biopsy or suction biopsy, we do not consider that this might have had any influence on the final diagnosis of non-HD or HD because of the cross checking between the PatoBank Registry and the National Patient Registry. For the non-HD cohort, the average age at inclusion (biopsy date) was significantly higher compared to the HD-group. An explanation might be that most patients in this group were investigated due to persistent constipation beyond the neonatal period.

A key limitation is that the study design did not allow for access to individual patient records to assess clinical history or indications for biopsy. This is an important factor that should be addressed in future prospective studies to ensure more accurate interpretation of diagnostic pathways.

## 5. Conclusions

Children with HD-negative rectal biopsies exhibited a significantly higher rate of abdominal surgeries, endoscopies, and imaging compared to the background population. This elevated healthcare utilization likely reflects ongoing or complex gastrointestinal symptoms. These findings underscore the need for structured diagnostic pathways, standardized management strategies, and closer follow-up to optimize care and resource use for children who had undergone rectal biopsy on the suspicion of Hirschsprung’s disease and without confirming the diagnosis. Further research is warranted to clarify underlying drivers of healthcare utilization.

## Figures and Tables

**Figure 1 children-12-01112-f001:**
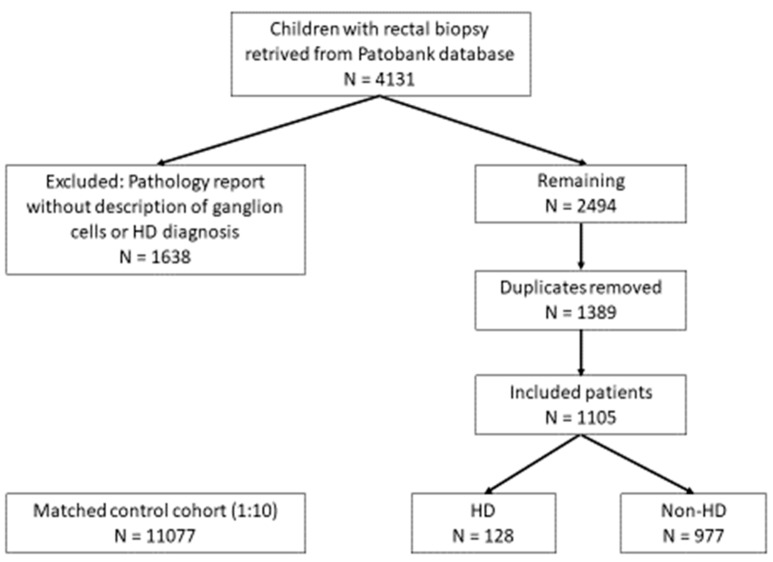
Flow diagram of patient inclusion.

**Table 1 children-12-01112-t001:** Baseline characteristics for three study cohorts: HD-group (Hirschsprungs’s disease), non-HD and healthy control group (no rectal biopsy). Values are given as means (standard deviation) for continuous variables and percentages for categorical data.

Variable	HD (N = 128)	Non-HD (N = 977)	Control (N = 11,077)	*p*-Value (HD vs. Non-HD)	*p*-Value (Non-HD vs. Control)
Age at biopsy (years)	0.9 (2.0)	2.8 (3.2)	2.6 (3.2)	<0.001	0.108
Follow-up (years)	6.8 (3.9)	5.6 (3.8)	5.8 (3.9)	0.002	0.150
Gestational age at birth (days)	272.9 (19.4)	269.8 (22.5)	277.5 (13.3)	0.136	<0.001
Length at birth (cm)	49.8 (9.5)	49.1 (9.2)	51.3 (5.6)	0.430	<0.001
Mother’s BMI	23.9 (3.9)	24.9 (5.8)	24.5 (7.4)	0.069	0.124
APGAR score (5 min)	9.1 (0.64)	9.1 (0.73)	9.86 (0.61)	0.932	0.002
Equivalated family income (EUR)	30,461 (17,850)	28,984 (14,496)	29,852 (18,089)	0.293	0.146
Mother’s smoking status					
Not active	107 (83.6%)	774 (79.2)	9272 (83.7%)		
Active	21 (16.4%)	203 (20.8%)	1805 (16.3%)	0.247	<0.001
Sex					
Female	29 (22.7%)	487 (49.8%)	5172 (46.7%)	<0.001	0.058
Male	99 (77.3%)	490 (50.2%)	5905 (53.3%)		

**Table 2 children-12-01112-t002:** Surgical procedures, imaging and endoscopies before and after index date (ID) in the group of patients with Hirschsprung’s disease (HD), non-HD and controls.

Procedure	HD (N = 128)	Non-HD (N = 977)	Control (N = 11,077)	*p*-Value (HD vs. Non-HD)	*p*-Value (Non-HD vs. Control)
Surgery before ID	34 (26.6%)	158 (16.2%)	131 (1.2%)	0.004	<0.001
Surgery after ID	128 (100.0%)	109 (11.2%)	175 (1.6%)	<0.001	<0.001
Number of surgeries after ID					
0	0	868 (88.9%)	10,902 (98.4%)	<0.001	<0.001
1–3	71 (56.8%)	91 (9.3%)	162 (1.5%)		
4–7	38 (30.4%)	11 (1.1%)	13 (0.1%)		
8+	16 (12.8%)	5 (0.5%)	0 (0.0%)		
Endoscopy before ID	5 (3.9%)	61 (6.2%)	33 (0.3%)	0.294	<0.001
Endoscopy after ID	52 (40.6%)	92 (9.4%)	57 (0.5%)	<0.001	<0.001
Any imaging before ID	121 (94.5%)	644 (65.9%)	550 (5.0%)	<0.001	<0.001
Any imaging after ID	78 (60.9%)	362 (37.1%)	862 (7.8%)	<0.001	<0.001
US before ID	36 (28.1%)	378 (38.7%)	419 (3.8%)	0.020	<0.001
US after ID	42 (32.8%)	277 (28.4%)	743 (6.7%)	0.295	<0.001
X-ray before ID	116 (90.6%)	503 (51.5%)	179 (1.6%)	<0.001	<0.001
X-ray after ID	69 (53.9%)	187 (19.1%)	142 (1.3%)	<0.001	<0.001
CT before ID	5 (3.9%)	20 (2.0%)	12 (0.1%)	0.183	<0.001
CT after ID	8 (6.2%)	17 (1.7%)	29 (0.3%)	0.001	<0.001
MRI before ID	0 (0.0%)	12 (1.2%)	4 (0.0%)	0.207	<0.001
MRI after ID	4 (3.1%)	21 (2.1%)	22 (0.2%)	0.485	<0.001

**Table 3 children-12-01112-t003:** Number and fraction (%) of the different surgical procedures performed after the index date in the three groups classified by NSCP (Nordic Classification of Surgical Procedures) codes.

NSCP Code	Procedure	HD (N = 128)	Non-HD (N = 977)	Control (N = 11,077)
KJF	Small and large intestine	99 (16.2%)	91 (39.4%)	7 (2.6%)
KJA	Abdominal wall	28 (4.6%)	48 (20.8%)	159 (58.7%)
KJH	Anus	326 (53.4%)	33 (14.3%)	13 (4.8%)
KJD	Stomach and duodenum	1 (0.2%)	26 (11.3%)	19 (7.0%)
KJG	Rectum	148 (24.3%)	11 (4.8%)	3 (1.1%)
KJC	Esophagus	-	10 (4.3%)	5 (1.9%)
KJE	Appendix	71 (0.2%)	7 (3.0%)	59 (21.8%)

## Data Availability

The original contributions presented in this study are included in the article/Appendix A. Further inquiries can be directed to the corresponding author.

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
