# Peer review of "Increased Utilization of Abdominal Surgical Procedures, Endoscopy and Imaging After Negative Rectal Biopsies for Suspected Hirschsprung’s Disease: A Danish Nationwide Matched Cohort Study"

_children, 2025, doi:10.3390/children12091112_

Round 1
Reviewer 1 Report (New Reviewer)
Comments and Suggestions for Authors
The authors conclude that patients with constipation not due to HD have a higher incidence of gastrointestinal interventions. I think the findings are interesting .
The authors need to clarify their methods a little better and make sure that their flow chart is accurate. There are two different numbers mentioned in the flowchart and the manuscript (Is it 4131 or 4132?).
They started with a little over 4100 patients who underwent rectal biopsy. They excluded 1638 patients for HD diagnosis or non-identification of ganglion cells in the pathology. Can the authors explain if these were neonates only?
After removing of duplicates they ended up with 1105 patients of whom 128 had HD and 977 did not. More clinical details about this cohort is necessary. Are these patients who underwent biopsy for intractable constipation at a later date?
What is the basis for a 10:1 propensity matching? Why not a simpler 2:1?
Can the authors clarify if the GI procedures performed in the non-HD group (977) were emergent or elective? It will help inform the reader as to what one should expect in the management of this group of patients.
Can the authors be a little bit more specific with regards to the operations rather than just saying stomach, SI, or LI? What exactly was done.
Author Response
Thank you very much for taking the time to review our manuscript and for your valuable and constructive feedback. We sincerely appreciate your insights, which have helped us improve the clarity and quality of the manuscript. Please find our detailed responses to your comments below.
The authors conclude that patients with constipation not due to HD have a higher incidence of gastrointestinal interventions. I think the findings are interesting.
The authors need to clarify their methods a little better and make sure that their flow chart is accurate. There are two different numbers mentioned in the flowchart and the manuscript (Is it 4131 or 4132?).
Response: Thank you for the comment. It was a typing error in the manuscript. The correct number is 4131 as stated in the figure. Corrected in the revised version of the manuscript.
They started with a little over 4100 patients who underwent rectal biopsy. They excluded 1638 patients for HD diagnosis or non-identification of ganglion cells in the pathology. Can the authors explain if these were neonates only?
Response: Thank you for this relevant comment. We have rephrased the first paragraph of the Result section that it included all ages below 18 years.
After removing of duplicates they ended up with 1105 patients of whom 128 had HD and 977 did not. More clinical details about this cohort is necessary. Are these patients who underwent biopsy for intractable constipation at a later date?
Response: Patients with a confirmed HD were censored for analysis of later rectal biopsies. This has been explained in the revised version in section of methods.
For the non-HD cohort, the average age at inclusion (biopsy date) was significantly higher compared to the HD-group. An explanation might be that most patients in this group were investigated due to a persistent constipation beyond the neonatal period. We have included this in the Limitation section of the revised version.
What is the basis for a 10:1 propensity matching? Why not a simpler 2:1?
Response: The 10:1 matching was used because of the relative low number of exposed individuals and to increase statistical power. Has been added to the methodology section in the revised version.
Can the authors clarify if the GI procedures performed in the non-HD group (977) were emergent or elective? It will help inform the reader as to what one should expect in the management of this group of patients.
Response: We do agree that this could be an important information, but the registry of surgical procedure contains no information on whether it was acute or elective. This has been explained in Discussion section of the revised manuscript.
Can the authors be a little bit more specific with regards to the operations rather than just saying stomach, SI, or LI? What exactly was done.
Response: We do agree that this is an important question, and as stated in the discussion section this requires a meticulous review of the patient records to identify indications and outcome, which wasn’t an aim of the present study, and we had no approval to perform this.
We have added the following to the discussion section: Our search for surgical interventions was confined to the main groups as e.g. stomach, small or large bowels etc. because the number of the detailed procedures performed were very low, and where a statistical analysis would give no meaning.
Reviewer 2 Report (New Reviewer)
Comments and Suggestions for Authors
This Danish study evaluates diagnostic invastigations and treatments in patients with suspected HD, but without a definitive diagnosis.
Line 45-47: remove “HD is a developmental malformation…”
Line 65-69: “To our knowledge no studies have investigated the use of radiological investigations, endoscopic procedures, and surgical interventions related to the gastrointestinal tract in children who had undergone a rectal biopsy on the suspicion of HD without confirmed diagnosis. We hypothesized that these children would undergo more investigations and surgical procedures related to the gastrointestinal tract as compared to a matched control group”: this is not true! In the Literature you can find a lot of paper with investigation on children with suspicion of HD.
Author Response
Thank you very much for taking the time to review our manuscript and for your valuable and constructive feedback. We sincerely appreciate your insights, which have helped us improve the clarity and quality of the manuscript. Please find our detailed responses to your comments below.
This Danish study evaluates diagnostic invastigations and treatments in patients with suspected HD, but without a definitive diagnosis.
Line 45-47: remove “HD is a developmental malformation…”
Response: Thank you for the comment. We do agree that this quotation may be inappropriate and have deleted it accordingly.
Line 65-69: “To our knowledge no studies have investigated the use of radiological investigations, endoscopic procedures, and surgical interventions related to the gastrointestinal tract in children who had undergone a rectal biopsy on the suspicion of HD without confirmed diagnosis. We hypothesized that these children would undergo more investigations and surgical procedures related to the gastrointestinal tract as compared to a matched control group”: this is not true! In the Literature you can find a lot of paper with investigation on children with suspicion of HD.
Response: Thank you for the comment. It is correct that there are numerous publications on investigations on children with the suspicion of HD, but not when it comes to investigations after an eventual rectal biopsy which excluded HD. We have rephrased the sections as follows:
There are numerous papers published on investigations performed on children with suspicion of HD. To our knowledge no studies have investigated the use of radiological investigations, endoscopic procedures, and surgical interventions related to the gastrointestinal tract after the children have undergone a rectal biopsy on the suspicion of HD without confirmed diagnosis. We hypothesized that these children would undergo more investigations and surgical procedures related to the gastrointestinal tract as compared to a matched control group
Round 2
Reviewer 1 Report (New Reviewer)
Comments and Suggestions for Authors
The authors have satisfied my questions and manuscript can be accepted
Author Response
Thank you for your kind remarks.
Reviewer 2 Report (New Reviewer)
Comments and Suggestions for Authors
In the Literature there are many papers about investigation in children with negative
rectal byopsies. Some authors found a redudancy of the colon
Author Response
Comment: In the Literature there are many papers about investigation in children with negative rectal byopsies. Some authors found a redudancy of the colon.
Answer: You are completely right. We have added a comment on this in the discussion section line 272-274 in the revised version.
Text added: Colonic redundance is often reported in patients with chronic constipation without HD, but the clinical significance of this condition is unknown and evidence for surgical resection is non-existing.
This manuscript is a resubmission of an earlier submission. The following is a list of the peer review reports and author responses from that submission.
Round 1
Reviewer 1 Report
Comments and Suggestions for Authors
Thank you for inviting me to review the manuscript “Increased utilization of abdominal surgical procedures, endoscopy and Imaging after negative rectal biopsies for suspected Hirschsprung’s disease: A Danish nationwide matched cohort study” submitted to Children. I have read this article carefully. Here are some of the questions about the content of this article.
The authors conducted a nationwide matched cohort study of Danish patients with negative rectal biopsies for suspected Hirschsprung’s disease, compared to matched healthy controls and patients diagnosed with HD, and concluded that children with a HD-negative rectal biopsy had 5-10 fold increased frequency of gastrointestinal-related surgeries, endoscopies or imaging during the follow-up period compared to the background population. The results were easily foreseeable. The authors confirmed this assumption through a large-scale study.
Comment 1: Lines 105-106 “Matching included age at biopsy date, and region of residence on the index date.” Why is it necessary to consider the region when matching?
Comment 2: As pointed out in the study, “It is also unknown whether this trend may persist beyond this relative short observation period.” Can the authors provide the specific causes for these non-HD patients receiving other treatments after the biopsy, especially the proportion of those who underwent treatments because of persistent constipation? This is the focus of the follow-up for these patients.
Comment 3: Much of the discussion in the manuscript was based on conjecture and lacked support from specific data. For instance, the author believed that the lower proportion of non-HD patients undergoing appendicitis surgery was due to the Marlon procedure. Still, there was no data. The authors need to confirm and provide this information.
Author Response
Reviewer #1
Thank you very much for taking the time to review our manuscript and for your valuable and constructive feedback. We sincerely appreciate your insights, which have helped us improve the clarity and quality of the manuscript. Please find our detailed responses to your comments below. All corresponding changes made in the revised version is highlighted with red
Comment 1: Lines 105-106 “Matching included age at biopsy date, and region of residence on the index date.” Why is it necessary to consider the region when matching?
Response: Thank you for pointing this out. The reason to include region of residence in the matching was to avoid any regional differences in indications for biopsy and biopsy methods
We have added the following to the limitations on page 3:
The inclusion of region of residence was to account for potential regional differences the indication for biopsy and used method (full-thickness or suction) for rectal biopsy.
Comment 2: As pointed out in the study, “It is also unknown whether this trend may persist beyond this relative short observation period.” Can the authors provide the specific causes for these non-HD patients receiving other treatments after the biopsy, especially the proportion of those who underwent treatments because of persistent constipation? This is the focus of the follow-up for these patients.
Response: We fully agree with this observation and have emphasized in the Discussion section that registry-based data inherently have limitations. They offer valuable insights into treatment patterns, but they do not provide detailed information on the indications for the specific treatments. As such, registry data cannot replace the granularity of prospective clinical data when it comes to causality or individualized patient analysis.
On page 7 (Discussion) we have added:
Registry-based data inherently have limitations. They offer valuable insights into treatment patterns, but they do not provide detailed information on the indications for the specific treatments. As such, registry data cannot replace the granularity of prospective clinical data when it comes to causality or individualized patient analysis.
Comment 3: Much of the discussion in the manuscript was based on conjecture and lacked support from specific data. For instance, the author believed that the lower proportion of non-HD patients undergoing appendicitis surgery was due to the Marlon procedure. Still, there was no data. The authors need to confirm and provide this information.
Response: Thank you for pointing this out. We agree with the comment, acknowledge that registry data is limited to identifying overall trends, and cannot provide the detailed individual-level information that prospective studies offer. We have only registered surgeries involving the appendix, without specifying the exact type of procedure. Appendicostomy is registered under the code for (KJF) in Table 3.
Thus, this sentence on appendicostomy makes no sense and the following has been deleted:
The low rate of appendix-related surgeries in the non-HD group is remarkable as one would expect that this might be higher because of a higher use of appendicostomy for antegrade colonic enemas in the non-HD group as this is by far the most common surgical treatment for severe constipation in children[18]. on page 7
We have added the following text:
The procedure code specific for appendicostomy, which is an option for treating severe constipation [18], could not be retrieved from the dataset.
Reviewer 2 Report
Comments and Suggestions for Authors
The authors present a large cohort of children identified through a nationwide registry including all children under the age of 18 years old. They have collected an impressive number of participants who have undergone rectal biopsies. There are some very interesting results and I do have some follow-up questions.
#1. Is it possible to know if the biopsies were taken as full-thickness biopsies or are some suction biopsies?
#2. There was an exclusion of 1638 children with rectal biopsies but without description of ganglion cells. Do the authors know why these rectal biopsies were taken and is there not a risk of missed diagnosis of these children?
#3. I found the data regarding smoking status interesting as this influences the birth age and weight and thus the risk of preterm babies which have a higher risk of gastrointestinal symptoms. Was it possible to do a sub-analysis in this group?
#4. Did the control group also undergo biopsies? In Table 1 this is registered and it seems a bit confusing as that would indicate that there are two exposed control groups?
# The authors mention that there is a risk of missing HD patients as the diagnosis have been based on other investigations. Is there perhaps a pathology exame made on all resected bowel after surgery for Hirschpsurng’s disease and could this not identify missing patients in that case?
There are some small langue errors that should be corrected e.g. surgery in Table S2, serous in Limitations. I would also suggest to have same order of the groups in all of the Tables (HD, non-HD, controls).
Author Response
Reviewer #2
Thank you very much for taking the time to review our manuscript and for your valuable and constructive feedback. We sincerely appreciate your insights, which have helped us improve the clarity and quality of the manuscript. Please find our detailed responses to your comments below. All corresponding changes have been made in the revised manuscript and are highlighted in the re-submitted files.
Comment 1: Is it possible to know if the biopsies were taken as full-thickness biopsies or are some suction biopsies.
Response: Thank you for pointing this out, we agree that it may be an important consideration. The SNOMED codes in the Patobank do not distinguish between biopsy types (full-thickness vs. suction), nor does the coding in the National Patient Registry (NPR). It is unlikely that this limitation has affected our results, as we cross-checked the diagnosis between the Patobank and the NPR. Moreover, the type of biopsy should not influence the final pathological diagnosis of HD or non-HD,
We have added the following on page 2.
Danish children (age <18 years) who underwent any type of rectal biopsy on suspicion of HD from January 1, 1998 to December 31, 2018.
In the limitation section we have added the following on page 8:
Although we could not distinguish between full-thickness biopsy or suction biopsy we do not consider that this might have had any influence on the final diagnosis of non-HD or HD because of the cross checking between PatoBank registry and the National Patient Registry.
Comment 2: There was an exclusion of 1638 children with rectal biopsies but without description of ganglion cells. Do the authors know why these rectal biopsies were taken and is there not a risk of missed diagnosis of these children?
Response: We agree that this could be a concern. In such cases, we assumed that the biopsy had been performed for reasons other than severe constipation, as the pathology report did not include a specific description of the presence or absence of ganglion cells, which is an essential criterion in the histological evaluation of rectal biopsies performed on the suspicion of HD.
We have edited the method section page 3:
The identified pathology reports were manually reviewed to confirm the description of the presence or absence of ganglion cells which is an essential criterion in the histological evaluation of rectal biopsies performed on the suspicion of HD.
Comment 3: I found the data regarding smoking status interesting as this influences the birth age and weight and thus the risk of preterm babies which have a higher risk of gastrointestinal symptoms. Was it possible to do a sub-analysis in this group?
Response: We thank the reviewer for this insightful comment. In the existing analyses, maternal smoking was a significant predictor of surgery, endoscopy, and imaging, respectively but not when adjusting for gestational age, length at birth, and previous investigations/surgery. This suggests that the predictive association is likely mediated through the impact of maternal smoking on birth age and size. A reliable subanalysis was not possible due to relative low number of patients.
The following text has been added on page 8.
Although maternal smoking was significantly associated with increased rates of surgery, endoscopy, and imaging, it became insignificant after adjusting for gestational age, birth size, and prior procedures. A reliable sub-analysis was not possible to perform due to the low number of cases.
Comment 4: Did the control group also undergo biopsies? In Table 1 this is registered and it seems a bit confusing as that would indicate that there are two exposed control groups?
Response: Thank you for pointing this out. The control group in the study consists of unexposed (no biopsy) matched controls. Matching was done one age, sex and place of residence.
We have clarified this in the method section on page 3:
An unexposed cohort (control group) without any registered rectal biopsy was matched at a 10:1 ratio using data from the Danish Civil Registry [14].
And edited the legend to table 1:
Table 1. Baseline characteristics for three study cohorts: HD-group (Hirschsprungs´s disease), Non-HD and the matched,control group (no rectal biopsy). Values are given as mean (standard deviation) for continuous variables and percentages for categorical data.
Comment 5: The authors mention that there is a risk of missing HD patients as the diagnosis have been based on other investigations. Is there perhaps a pathology exame made on all resected bowel after surgery for Hirschpsurng’s disease and could this not identify missing patients in that case?
Response:
To be included in the HD-group required a diagnosis based on the histological examination of the rectal biopsy and that the patient was registered in the National Patient Registry with a diagnosis code of HD. Patients without a biopsy and diagnosed with HD would have been missed. We consider this scenario to be very low, below 1%. We did not perform any review of the surgical specimens in patients that appeared in the NPR with a HD diagnosis without a rectal biopsy, which could by another limitation.
This has been added to the limitation section on page 8:
We did not perform any review of the surgical specimens in patients that appeared in the NPR with a HD diagnosis without a rectal biopsy, which could by another limitation.
Comment 6: There are some small langue errors that should be corrected e.g. surgery in Table S2, serous in Limitations. I would also suggest to have same order of the groups in all of the Tables (HD, non-HD, controls).
Response: The language errors have been corrected accordingly, and Tables has been changed accordingly.
Reviewer 3 Report
Comments and Suggestions for Authors
The title and purpose of this study promise significant research on a very interesting and important topic.
Unfortunately, when looking at the content of the study, it appears to be a work prepared with computer data at the desk, and there is a lack of noteworthy information. This study is conducted on patients with functional constipation, but misleadingly, this situation is not mentioned in the title.
Some important information needs to be added for this study to be considered a noteworthy significant work.
According to our information, Hirschsprung's disease is one of the first diseases to consider in a patient presenting with functional constipation. There are 3 important procedures that need to be done before taking a rectal biopsy in a patient presenting with functional constipation.
1- Anamnesis (investigation of family history of Hirschsprung's disease - Was the first meconium passed within 24 hours after birth?)
2- Distal colon X-ray AP-LATERAL (Narrow segment - transition zone - dilated segment - normal segment)
3- Anorectal manometry results.
These evaluation methods should be performed and results should be positive for HD before any functional constipation patient undergoes a rectal biopsy.
Because the real important question is whether the rectal biopsies were taken with the correct indications for the patients.
Imagine a surgeon who takes a biopsy from every patient with constipation that comes to them and then performs surgery. What has been done has no contribution to the literature; in fact, it conveys incorrect information.
In this study, I want each patient's three parameters to be investigated and added (Stool passage in the first 24 hours postpartum in the anamnesis - Distal colon radiography - Anorectal manometry results). In other words, I request a major revision. Please do not add the absence of these findings to the limitations, as these parameters are not findings to be underestimated.
Author Response
Reviewer #3
Thank you very much for taking the time to review our manuscript and for your valuable and constructive feedback. We sincerely appreciate your insights, which have helped us improve the clarity and quality of the manuscript. Please find our detailed responses to your comments below. All corresponding changes have been made in the revised manuscript and are highlighted in the re-submitted files.
Comment 1: This study is conducted on patients with functional constipation, but misleadingly, this situation is not mentioned in the title.
Response: Thank you very much for pointing this out. We fully acknowledge your comment. Suspicion of Hirschsprung’s disease may arise from a range of clinical indications, varying from newborns with significant delay in passing meconium to older children presenting with symptoms of severe constipation. Given this clinical diversity, we chose not to specify the indications for rectal biopsy in the title to reflect the broad spectrum of presentations that can lead to diagnostic evaluation.
Comment 2: There are 3 important procedures that need to be done before taking a rectal biopsy in a patient presenting with functional constipation.
1- Anamnesis (investigation of family history of Hirschsprung's disease - Was the first meconium passed within 24 hours after birth?)
2- Distal colon X-ray AP-LATERAL (Narrow segment - transition zone - dilated segment - normal segment)
3- Anorectal manometry results.
These evaluation methods should be performed and results should be positive for HD before any functional constipation patient undergoes a rectal biopsy.
Because the real important question is whether the rectal biopsies were taken with the correct indications for the patients.
Response: Thank you very much for your thoughtful and important comment. We fully acknowledge the significance of a thorough diagnostic evaluation prior to performing a rectal biopsy.
Ad. 1: With the design of the present study, we were not able to review the patient records for history and indication for taking the biopsy. It is an important issue but will require prospective investigation to get reliable information.
We have added the following to the limitations on page 8:
A key limitation is that the study design did not allow access to individual patient records to assess clinical history or indications for biopsy. This is an important factor that should be addressed in future prospective studies to ensure more accurate interpretation of diagnostic pathways.
Ad. 2 and 3:
Anorectal manometry and contrast studies of the colon was not routinely used in diagnostic work-up for Hirschsprung’s disease in Denmark during this period and afterwards. Rectal biopsy is the primary investigation.
We have added this information in the discussion, Page 7:
Anorectal manometry and contrast studies were not routinely part of the diagnostic work-up for Hirschsprung’s disease in Denmark during the study period or thereafter. Diagnosis relied primarily on rectal biopsy.
Round 2
Reviewer 1 Report
Comments and Suggestions for Authors
Accept in present form.
Author Response
Thank you very much for the acceptance of the revision of the manuscript. We very much appreciate your efforts.
Reviewer 2 Report
Comments and Suggestions for Authors
Thank you for clarifying the paper accordingly to the last review. I have no further questions or concerns.
Author Response
Thank you very much for your acceptance of the revised manuscript. Your efforts are very much appreciated.
Reviewer 3 Report
Comments and Suggestions for Authors
There are very serious structural errors in the design of this study. It is incorrect to perform a rectal biopsy on a patient with functional constipation without evaluation of clinical presentation, a radiological evaluation and anorectal manometry (1). A surgeon should perform the correct surgical intervention with the right indication. There is no evidence of a correct indication in this study. If this study is published in an important journal, it will set a precedent for taking rectal biopsies without any prior examinations following the referral of all functional constipation patients. It is more appropriate for a surgeon to take a rectal biopsy if there is a strong suspicion of Hirschsprung disease (1). Because this protects the patient from unnecessary surgical intervention and anesthesia risk. However, in this study, this was not done; approximately 60-70% of the patients are not Hirschsprung. Also, the differential diagnosis is important. You cannot diagnose microcolon or congenital pouch colon with a rectal biopsy without a radiological contrast study, because in these diseases, ganglion cells are positive. I recommended to the authors to correct the errors in the study, but the authors chose to convince me that there is no significant flaw in the study. I do not accept this (1).
1-Teitelbaum DH, Coran AG. Hirschsprung’s disease and related neuromuscular disorders of the intestine. In Grosfeld JL, O’Neil JA, Coran AG, Fonkalsrud EW. Textbook of Pediatric Surgery. 6th Ed. Philadelphia: 2006; 2(99):1514-59.
Author Response
We fully understand and appreciate your concerns on the indication to perform a rectal biopsy in children on the suspicion of Hirschsprung´s disease, but this was not an objective of the present study as explained very clearly in the manuscript.
Rectal biopsy is the gold standard to diagnose or exclude HD in a child with severe constipation or strained defaecation. It is correct that most patients will undergo some kind of imaging including barium enema, but the sensitivity and specificity is a problem. In a study form 2007 the rate of false positive CE vas 48% in patients undergoing rectal biopsy on the suspicion on HD (J. Pedr Surg 20027;42(5):792-5). We acknowledge that the false negative results are lower but may have severe consequences for the child. The same applies to the use of anorectal manometry. The Nice Guideline for Constipation in children and young people; diagnosis and management recommend that you should not use anorectal manometry to exclude HD or plain abdominal X-ray. There is no recommendation on colonic enema. The guidelines recommend that the decision on rectal biopsy on the suspicion of HD should be on at detailed history and physical examination.
In our register-based study we found a relative high incidence of imaging prior to rectal biopsy, which was similar for HD and non-HD groups, which indicate that imaging is common in in patients with severe constipation, but we cannot give any detailed information on the type of imaging and the results. In the discussion section under the 2nd paragraph, we have added the following text: ……during the study period or thereafter in line with the NICE recommendations [2]. The high incidence of imaging procedures prior to index date (biopsy) found in both the HD and non-HD group may reflect a widespread use of diagnostic imaging prior to the decision of biopsy, but we have no information on the results of these investigations.
But all this is fundamental other discussion and can only be answered in controlled cohort studies and not in a registry study as ours. We investigated what happens later in life with those patients who had a negative biopsy for HD. The results are to our opinion important a surprisingly with the finding of a high utilization of imaging and a high number of gastrointestinal surgeries which calls for further analysis to improve the medical care of these vulnerable patients.
We understand your concerns about the indications for rectal biopsy, but this was not an objective in the present study. By your focus upon this concern only, we find that your assessment of the manuscript perhaps not is based on the right premises, but this must be up to the Editor to decide.